# Evaluation of Deformable Image Registration under Alignment-Regularity Trade-off

**Abstract.** Evaluating deformable image registration (DIR) is challenging due to the inherent trade-off between achieving high alignment accuracy and maintaining deformation regularity. However, most existing DIR works either address this trade-off inadequately or overlook it altogether. In this paper, we highlight the issues with existing practices and propose an evaluation scheme that captures the trade-off continuously to holistically evaluate DIR methods. We first introduce the alignment-regularity characteristic (ARC) curves, which describe the performance of a given registration method as a spectrum under various degrees of regularity. We demonstrate that the ARC curves reveal unique insights that are not evident from existing evaluation practices, using experiments on representative deep learning DIR methods with various network architectures and transformation models. We further adopt a HyperNetwork-based approach that learns to continuously interpolate across the full regularization range, accelerating the construction and improving the sample density of ARC curves. Finally, we provide general guidelines for a nuanced model evaluation and selection using our evaluation scheme for both practitioners and registration researchers. [1]

**Keywords:** Image registration · Deformable Registration · Evaluation

## 1 Introduction

Image registration is one of the most fundamental tasks in medical imaging and analysis. The core aim of image registration is to find spatial transformations that align anatomical structures or functional elements across two or multiple images. This is often achieved by automatically adjusting a transformation model to optimize the *alignment* according to a predefined dissimilarity measure. In deformable image registration, the transformation model is allowed high degrees of freedom, which warrants additional regularization to enforce desirable properties, or *regularity*, and constrains the solution space for more efficient optimization. Concretely, many deformable image registration algorithms, including iterative

---

[1] Code is available at: https://anonymous.4open.science/r/arc-3F80

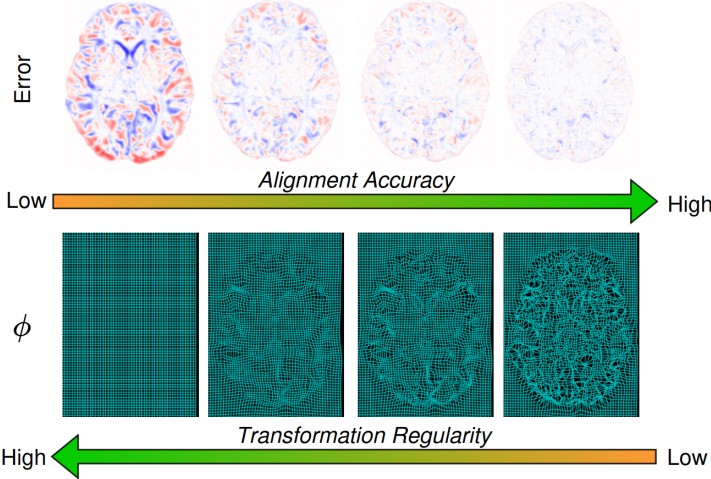

Fig. 1: Illustration of the balance between *alignment* accuracy and transformation *regularity* during inter-subject brain registration.

optimization registration and modern learning-based registration methods, solve the following energy optimization problem to register a pair of images [17,19]:

$$\tilde{\phi} = \arg\min_{\phi}[\mathcal{D}(T(I_m, \phi), I_f) + \lambda\mathcal{R}(\phi)] \tag{1}$$

Here the dissimilarity term $\mathcal{D}$ quantifies the alignment between the *moving* image $I_m$ transformed by $T$ using the transformation $\phi$ (*moved*), and the reference image $I_f$. The transformation is constrained by the regularization term $\mathcal{R}$, which is usually derived to enforce desirable properties on the transformation, such as smoothness or topology-preservation. This governs the regularity of the transformation. The weighting hyperparameter $\lambda$ in Eq. (1) influences the balance between the two energy terms. The resulting dynamic is a trade-off between the accuracy of the alignment and the regularity of the transformation. A visual illustration of this trade-off is shown in Fig. 1. Formally, we term this trade-off as the alignment-regularity characteristic (ARC) of a given registration algorithm, inspired by the use of receiver operating characteristic (ROC) curves [3] in evaluating classification methods under the precision-recall trade-off.

**Issues with current practices:** Evaluating DIR algorithms in the context of ARC is not trivial. We observe that many registration works do not consider the ARC trade-off when evaluating and comparing results. For example, many methods only tune $\lambda$ to maximize alignment accuracy. We argue that this is problematic in a few ways: *1) Lack of controlled comparison:* Performance between methods is often compared without controlling for alignment or regularity. For example, many works in learning-based deformable image registration (LDIR) regard the optimal hyperparameter $\lambda$ value to be the one that maximizes anatomical alignment (e.g., Dice score) [2,13,4]. Therefore, alignment results are

often reported with non-comparable regularity of the transformation. As we demonstrate later, this can lead to misleading or ambiguous conclusions since it is unclear whether a higher degree of alignment at the expense of regularity is preferable. *2) Discrete-points bias:* Most existing DIR works in the literature report and compare results at discrete points on the alignment-regularity trade-off spectrum [5,8] (with the only limited exception found in [10]). However, we found that LDIR methods often exhibit different relative performances at different levels of the regularity. This renders comparison on discrete points incomplete even if alignment or regularity are controlled to be comparable, since different conclusions can be drawn at different points of the spectrum. Moreover, finding comparable discrete points for evaluation can be challenging. The parameters and configurations of the algorithms usually do not control the metric values precisely and continuously due to the stochastic nature of the optimization process. As demonstrated later, adopting a more continuous comparison scheme could help mitigate this issue. *3) Ignoring application-dependent preferences:* An incomplete evaluation of the alignment and regularity trade-off omits crucial information since the desired registration algorithm properties are often application-dependent. For example, atlas-based segmentation may tolerate topological changes to improve structural matching and label propagation, while applications such as multi-modal fusion or respiratory motion tracking expect the transformation to be well-behaved and topology-preserving. Providing performance evaluation in a wider range of settings provides the users with more information enabling them to select the optimal algorithm for their applications.

**Contributions:** To address the issues mentioned above, we introduce an evaluation scheme that examines the alignment-regularity characteristic of DIR algorithms holistically to better inform model evaluation and selection. We focus on deep learning methods that utilize the optimization objective in Eq. 1, although our evaluation scheme is not limited to learning-based methods. Our contributions are summarized as follows:

1. We propose the construction of ARC curves based on alignment accuracy and deformation regularity metrics, demonstrating that these curves provide valuable and unique insights for method evaluation and comparison.
2. We employ a HyperNetwork-based approach that learns a continuous functional mapping between the regularization hyperparameters to the registration networks parameters, as a model-agnostic solution to accelerate ARC curve construction.
3. We demonstrate our evaluation scheme on representative methods and two widely-used datasets from the Learn2Reg challenge, namely the MRI brain dataset OASIS and the CT lung dataset NLST.

## 2 Alignment-Regularity Characteristics Curve

**Method:** To construct the alignment-regularity characteristic (ARC) curve for a given registration algorithm and dataset, we perform registration using varying

levels of regularization by varying the weighting $\lambda$ in Eq. 1. The ARC curve is then generated by aggregating the metric measurements across the test dataset and plotting the accuracy metric against the regularity metric. Crucially, we use the regularity metric instead of the regularization weight to normalize across the variation of loss formulation and implementation between methods. Examples of these curves are shown in Figure 2. In the following sections, we show empirically that valuable insights and comprehensive performance evaluation can be obtained by comparatively analyzing different methods using ARC curves.

**Experimental settings:** We acquire registration results and construct ARC curves using a range of different registration methods and two distinctive datasets.

- *Network Architectures:* We trained and evaluated several well-studied and state-of-the-art methods that demonstrate different architectural characteristics. We include VoxelMorph [2] and TransMorph [4], which are single-resolution models learning non-parametric dense deformations using a U-Net [16] and a Swin-Transformer [11]-based architecture, respectively. We also include two representative methods that focus on multi-resolution (LapIRN [13]) or multi-cascade (RCN [21]) refinement through composition.
- *Transformation models:* To study the effect of transformation models on ARC, we include MIDIR [15] which learns a parametric transformation model based on control points (free-form deformation or FFD [18]), as well as variants of all the aforementioned architectures that predict the stationary velocity field (SVF) [1] for diffeomorphic large deformation. We set the number of Scaling-and-Squaring integration steps to 7 for all SVF models. Non-parametric displacement field methods are denoted by "Disp".
- *Training strategy:* To obtain the ARC spectrum, we trained each method with a set of regularization weights $\lambda = (0.0, 0.001, 0.005, 0.1, 0.2, 0.5, 1.0)$ for 300 epochs each, using the ADAM [9] optimizer with a learning rate of $10^{-4}$ and a batch size of 2. We use negative normalized cross-correlation (NCC) as the dissimilarity term ($\mathcal{D}$) and the diffusion regularizer [7] ($\mathcal{R}$).
- *Datasets:* We perform our experiments on two widely-benchmarked datasets from the Learn2Reg challenge [5]. With the OASIS [12] dataset, we construct inter-subject registration pairs out of the T1-weighted brain MR images and 4-label segmentations[2] of 394 subjects for training/validation, and 20 subjects for testing. From the NLST [20] dataset, we use 150 pairs of inhale-exhale CT scans with lung masks and automatically detected landmarks for intra-subject registration, with a 90%-10% train/val-test split.
- *Evaluation metrics:* Due to the lack of ground truth transformation, evaluations of alignment accuracy in DIR are usually measured with surrogate metrics. For OASIS, we evaluate the *alignment* by measuring the overlap between the segmentation labels of the fixed scan and the warped moving scan via Dice score. For NLST, we utilize the available anatomical landmarks and evaluate the Target Registration Error (TRE), which measures the Euclidean distance between registered landmarks. The transformation *regularity* is evaluated by both the percentage of grid points with a negative

---

[2] https://github.com/adalca/medical-datasets/blob/master/neurite-oasis.md

Jacobian determinant (folding ratio), which is a proxy metric for topological changes, and the standard deviation of the logarithm of the Jacobian determinant (stdLogJ)[5], which indicates deformation smoothness.

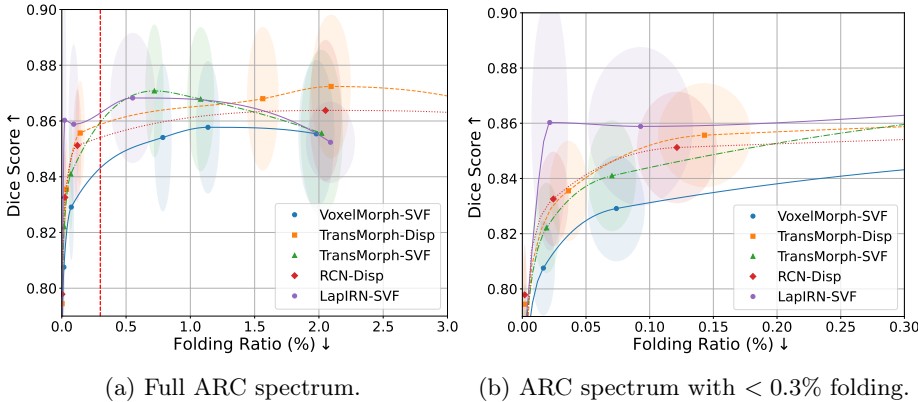

(a) Full ARC spectrum.          (b) ARC spectrum with $< 0.3\%$ folding.

Fig. 2: Full spectrum (a) and low folding ratio regime of the same spectrum (b) of the ARC curves for representative methods for inter-subject brain registration on OASIS dataset. The red dashed line on (a) indicates a 0.3% folding ratio. The shaded ellipse around each data point indicates the standard deviation of the metrics across the test pairs.

**Insight 1. Holistic model comparison using ARC curves:** Existing learning-based deformable image registration methods often tune the regularization level to optimize an alignment accuracy metric, such as the Dice score, without considering whether or not the transformation regularity at these operating points is comparable with competing methods [2,13]. The issue with this approach is exposed when we compare a few methods using the proposed ARC evaluation, as we empirically demonstrate with ARC curves constructed from a range of methods using the OASIS dataset shown in Fig. 2.

*Firstly*, we can see that different methods exhibit optimal Dice scores with distinctively different folding ratios. This means simply comparing the maximum Dice score omits the differences in regularity and their consequences in different applications. For example, we can see from Fig. 2a that the TransMorph-SVF, TransMorph-Dips, and RCN-Disp all outperform LapIRN-SVF in terms of maximal Dice score, but at the cost of much higher ($\sim 6\times$) folding ratio. Therefore, one cannot conclude that LapIRN-SVF performs inferior compared to the other two methods without considering if this higher deformation irregularity is acceptable in their specific application. For applications that require well-regularized transformations (i.e. lower folding), the LapIRN-SVF model should be preferred. We highlight this by focusing on the low-folding regime of the ARC curves shown in Fig. 2b. *Secondly*, while existing works report relative

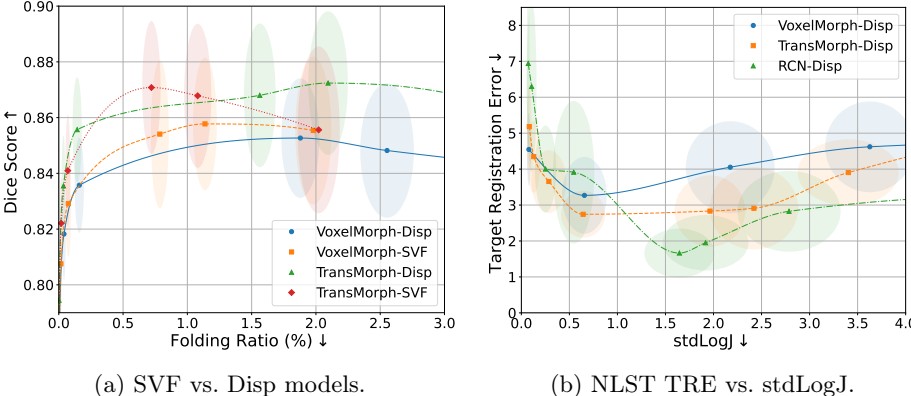

(a) SVF vs. Disp models.                    (b) NLST TRE vs. stdLogJ.

Fig. 3: ARC curves demonstrating (a) the effect of the SVF vs Disp transformation models on the performance under varying levels of regularization and (b) different metrics using the NLST dataset.

performances with comparable but discrete points of alignment accuracy and regularity (numbers in tables), the effectiveness of such comparisons is hindered by the choice of where the values are compared on the spectrum. Different conclusions can be drawn from "slicing" at different points on the ARC spectrum, as made evident by comparing Fig. 2a with Fig. 2b. Therefore, a continuous representation of performances is necessary to compare models comprehensively across the entire regularity spectrum. Similar conclusions can be reached when examining ARC curves with landmark-based TRE as alignment accuracy and Jacobian variation as regularity on the NLST dataset, as shown in Fig. 3b.

**Insight 2. Influence of transformation models:** In this section, we examine the effect of the SVF model on the performance of LDIR methods using ARC curves. SVF is a diffeomorphic transformation model with good theoretical regularity while capable of modeling large deformations. We noticed that recent SVF-based models reported in registration literature often under-perform models using simple displacement fields with the same network architecture [4,13]. However, from our experiments shown in Fig. 3a, we observe that SVF methods are only less accurate than displacement methods in the extreme low-folding regime. Both SVF models show higher accuracy than their displacement counterparts when allowed a slightly higher folding ratio (e.g. around 1%, which is still significantly lower than the displacement models at optimal Dice). We hypothesize that researchers often apply higher regularization on the velocity fields to enforce zero folding since SVF models are expected to be diffeomorphic, resulting in lower accuracy and narrowly missing the optimal operating points. In addition, we found that the SVF models are more robust to lower regularization weights, especially in the lower folding ratio zone. These insights are valuable to model selection and are only revealed through the ARC curves.

## 3   Amortized alignment-regularity characteristic

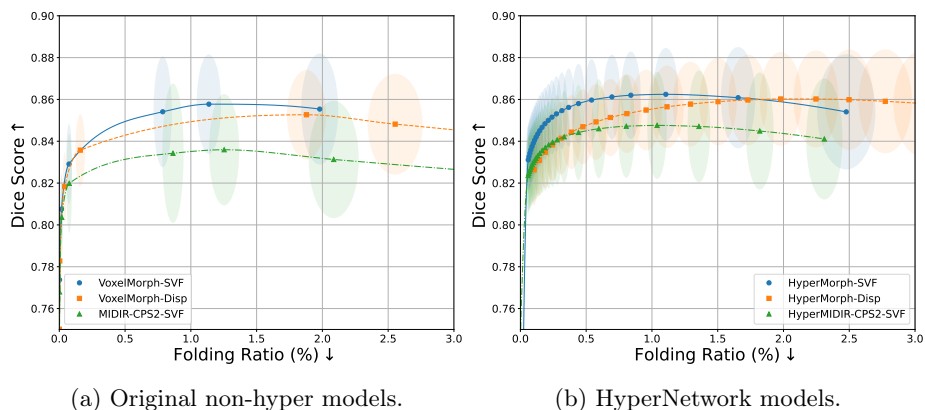

(a) Original non-hyper models.        (b) HyperNetwork models.

Fig. 4: ARC curves demonstrating the ARC spectrum of the (a) original versus the (b) HyperNetwork SVF and Disp models.

Despite showing promise in improving evaluation, ARC curves constructed using the method introduced in Section 2 have two main drawbacks. First, acquiring the curves is computationally expensive as each point on an ARC curve is a training-testing instance of a LDIR model using a specific regularization weight. This contributes to the second limitation, which is the sparsity of the data points. This necessitates post-hoc interpolation to approximate a smooth spectrum, which hinders the completeness of ARC curve-based evaluation as data points usually do not cover the whole spectrum evenly, as exemplified by Fig. 4a.

To address these problems, we leverage the hyperparameter amortization framework from HyperMorph [6] to enable fast and dense sampling of ARC data points without having to train a model for each regularization weighting $\lambda$. This is facilitated by using HyperNetwork, which is a network that learns a continuous functional mapping between regularization weights to the parameters of the registration networks. Consequently, we can sample the model regularity at arbitrary points on the spectrum at test time significantly more densely than conventional hyperparameter tuning without additional training. We favor HyperMoprh over the conceptually similar conditional LapIRN [14] framework as the former theoretically supports any network architecture. To meaningfully utilize this system for our ARC evaluation scheme, we experiment with using this framework to capture the regularization characteristics of different network architectures and transformation models. Specifically, we trained HyperNets to amortize $\lambda$ for VoxelMorph but with a different transformation model (SVF) and MIDIR [15], which has a different network architecture adapted for FFD.

The results of these experiments are presented in Figure 4b, where the HyperNetworks are shown to be capable of capturing relative performances between different network architectures and transformation models. More advantageously, these hyper-models are able to sample the regularization weight $\lambda$ in a more continuous manner across the entire ARC spectrum without incurring additional training time. This provides more detailed information for model evaluation and selection in a computationally efficient way at test time.

## 4   Conclusion and Discussion

**General guideline for model evaluation and selection:** *For researchers* working on DIR algorithms, we advocate for reporting ARC curves of any proposed methods and baseline methods to enable a fairer assessment of the contribution and provide a performance profile for downstream users. This could also clarify research directions for the registration community. *For practitioners* applying registration algorithms, we recommend constructing ARC curves for candidate algorithms with various configurations using a small subset of the data. The application-dependent optimal range of regularity can be identified by qualitatively examining the deformation field and deformed images at varying levels of regularity. Then, the best algorithm can be chosen according to the performances and trends in the optimal/acceptable regularity range on the ARC curve. The sensitivity of the competing algorithms to the regularization hyperparameters can also be assessed via ARC curves, which can be critical in selecting the optimal solution. In both cases, the amortized ARC system (Section 3) can also be utilized to accelerate the evaluation process if the additional computational cost can be afforded.

**Limitations:** The studies in this work should be expanded to more datasets, algorithms, and evaluation metrics such as intensity-based metrics for alignment and non-Jacobian-based metrics for regularity. A single-value metric, such as area under the ARC curves (AUC-ARC) similar to AUC-ROC, can provide a simpler solution to the evaluation problem. However, the exact formulation of AUC-ARC is not trivial since the theoretical bounds are empirically difficult to achieve. We are actively working on a solution and will present the results in a future work.

**Conclusion:** This work highlights the issues in current evaluation practices of deformable image registration under the alignment-regularity trade-off, and proposes an evaluation scheme using alignment-regularity characteristic curves to address these issues. We demonstrated the utility of such an evaluation system through several unique and valuable insights gained from applying the system, along with accelerated HyperNetwork-based variants. Finally, we provide general guidelines for researchers and practitioners on using the ARC scheme to evaluate and select deformable image registration methods.

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
