# OpenReview forum: "Evaluation of Deformable Image Registration under Alignment-Regularity Trade-off"
_MICCAI.org/2025/Workshop/BRIDGE — BRIDGE 2025 Poster_

### Official Review · Reviewer_x6T8 · 2025-07-24
**Review Comments**

**Rating:** 7
**Confidence:** 4

**Review:**

You have addressed an important gap in the evaluation of deformable image registration (DIR), that is, the balance between alignment accuracy and transformation regularity. Introducing the Alignment–Regularity Characteristic (ARC) curves is a valuable idea. Plotting accuracy against a regularity metric gives a convincing view of how different methods trade off these objectives. That said, I have a few concerns regarding your experimental methodology.
1.	The choice and interpretation of regularity metrics need justification. Why use folding ratio and the standard deviation of log‐Jacobian rather than, say, Hausdorff distances on deformation fields or measures of smoothness like bending energy? A brief experiment or literature reference showing that these two metrics capture the most clinically relevant aspects of regularity would strengthen the argument.
2.	I am left wanting more implementation details for reproducibility. The paper lists λ values and training schedules for standard models, but the Hyper-Network setup is described superficially. Please specify the architecture, how λ is encoded as input, what loss terms ensure fidelity at each operating point, and which random seeds or code versions you used. Publishing a GitHub link or supplementary code would allow reproducibility.
3.	The experiments cover two public datasets and a range of models. To show broad utility, I suggest adding one more scenario, perhaps a small clinical dataset or a different anatomy (e.g., abdominal CT), or at least demonstrating ARC curves on a publicly available, non‐Learn2Reg benchmark. That would confirm that ARC insights generalize beyond the two cases shown.
4.	The presentation could be tightened. All symbols (e.g., D, R, λ, ARC) belong in one “Notation” subsection. Figure captions should include sample sizes, kernel details for NCC loss, and hyperparameters.
5.	In Tables or plots, adding shaded error bars or mean±std would help me judge whether observed differences are statistically meaningful.
6.	A short discussion could acknowledge that ARC curves currently ignore computation time, that Hyper-Networks add overhead at training time, and that evaluating topology preservation beyond fold percentage remains an open problem.

---

### Official Review · Reviewer_Ljga · 2025-07-25
**Reviewer's Comments**

**Rating:** 9
**Confidence:** 4

**Review:**

### 1. Summary of the Paper
This paper presents a new way to evaluate medical image registration methods by looking at the trade-off between alignment accuracy and deformation regularity. The authors create what they call an Alignment-Regularity Characteristic curve as a continuous, holistic evaluation method. The authors test this evaluation approach on multiple state-of-the-art registration methods using benchmark datasets to show its value.

### 2. Strengths
* The ARC curve provides a comprehensive and insightful way to assess DIR performance
* The study includes diverse architectures, transformation models, and datasets, with consistent evidence supporting the utility of the proposed evaluation approach.

### 3. Limitations or Areas for Improvement
* Reporting label-wise Dice or class-wise TRE to understand substructure variance would be interesting. Understanding how performance varies across anatomical labels or substructures could provide further insights.

### 4. Relevance to BRIDGE Workshop Topics
The ARC framework directly tackles how registration methods perform under different constraint levels, which is key for understanding how reliable and generalizable these models are in practice.

---

### Official Review · Reviewer_2zJh · 2025-07-25
**This kind of evaluation aligns very closely with regulatory needs; nice work!**

**Rating:** 9
**Confidence:** 4

**Review:**

This paper identifies and addresses a current gap in deformable image registration evaluation, which is the trade‑off between alignment accuracy and transformation regularity. It presents the ARC curves, which continuously map registration performance across varying regularization strengths. This is a very nice paper, and I am genuinely looking forward to seeing the extension of this work.

Paper’s strengths:

1. First, from a regulatory science perspective, this work aligns closely with the regulatory need to understand a model’s behavior across its entire operating range to better assess risk and safety, as relying on a single metric or evaluation threshold does not provide the full image or capture how performance shifts as regularization varies. The paper is really nice as it covers three important regulatory needs when it comes to evaluation:
	a. Relying on a specific metric or evaluation aspect is not enough; we should consider other important aspects of the tasks,
	b. evaluation criteria change depending on the application and its intended use,
	c. It is always good to understand the performance of a model across the full spectrum of thresholds, and it is not enough to report performance at one threshold. The analysis of performance across different metrics and thresholds facilitates the understanding of the risk and failure of these methods, which is important from a regulatory perspective.

The analysis of ARC curves expose how different registration methods navigate the full spectrum of flexibility versus smoothness, rather than relying on single‑point comparisons. This is clearly demonstrated in Figure 2.

2. The idea of embedding regularization weight as a continuous input to a HyperNetwork is neat, as it allows the framework to avoid retraining a separate model for each λ, enabling dense, computationally efficient exploration of the alignment–regularity frontier.

3. The discussion  in the conclusion on general guidelines for model evaluation and selection is a very valuable step toward standardized frameworks.

Paper’s Weaknesses (more like suggestions/comments):

1. One thing to think about is that if ARC is computed over a regularization range not centered on clinically relevant λ values, methods tuned to “easy” regions can dominate the evalaution, and mask poor performance where it matters most. Can you please clarify if this is the case?
2. Does disproportionate sampling around a population‑optimal region artificially inflate ARC?
3. If ARC metrics are computed globally over the entire image, how does this impact local and more critical regions (e.g., near lesions or critical structures)?
4. Does averaging ARC curves across subjects individual “sweet‑spots,” risking suboptimal λ choices for outlier anatomies that deviate from the mean?
5. It will be useful to see how better evaluation can impact downstream tasks, and translate into clinical impact.

This work is very suitable for the BRIDGE workshop!!

---

### Decision · Program_Chairs · 2025-07-25

**Decision:**

Accept (Poster)

**Comment:**

Dear Authors,

Congratulations!

We are delighted to inform you that your paper has been accepted for publication in the BRIDGE Workshop. Your submission underwent rigorous peer review by three experts representing regulatory, academic, and industry perspectives, and their comments are provided below.

Requirements for your final camera‑ready submission (due July 30):
* Incorporate reviewer comments and suggestions throughout your paper; at minimum, add a discussion section addressing the key points raised by reviewers.
* Ensure your final draft follows standard MICCAI conference and Springer formats and guidelines.
* Submit your camera‑ready source file and any supplementary material.

We look forward to your presentation and the discussions it will generate at the workshop!

Best regards,
BRIDGE Workshop Organizers